# AIMCoT: Active Information-driven Multi-modal Chain-of-Thought for Vision-Language Reasoning

## Abstract

Multimodal Chain-of-Thought (CoT) has emerged as a powerful technique for enhancing the vision-language reasoning with interleaved information. However, existing methods often rely on simplistic heuristics for constructing interleaved CoT, typically depending on attention maps, which our empirical analysis reveals can be unreliable. What's more, the shortcomings of their passive and purpose-less selection strategies and their arbitrary triggering mechanisms in capturing the model's cognitive need for information are further amplified. In this paper, we propose **AIMCoT**, an **A**ctive **I**nformation-driven **M**ulti-modal **C**hain-**o**f-**T**hought framework that addresses these fundamental limitations. AIMCoT introduces three synergistic components: (1) **Context-enhanced Attention-map Generation (CAG)**, which mitigates the text-vision granularity imbalance, thereby producing more reliable attention maps as a foundation. (2) **Active Visual Probing (AVP)**, which replaces passive selection with a proactive, goal-oriented strategy grounded in information theory to select image regions that help answer the questions maximally. (3) **Dynamic Attention-shifting Trigger (DAT)**, which intelligently determines the optimal moments to insert visual information by monitoring the model's text-to-vision attention shifts. Extensive experiments on three challenging benchmarks demonstrate that AIMCoT significantly outperforms state-of-the-art methods across different settings. By actively foraging for information and dynamically structuring its reasoning process, AIMCoT represents a critical step towards more robust, effective, and human-like multimodal reasoning. Our code is available at https://anonymous.4open.science/r/AIMCoT.

## 1 Introduction

The advent of Chain-of-Thought (CoT) prompting has marked a significant milestone in the reasoning capabilities of Large Language Models (LLMs) Wei et al. (2022); Wang et al. (2022); Zhang et al. (2022); Suzgun et al. (2022); Li et al. (2025); Yao et al. (2023); Besta et al. (2024), enabling them to deconstruct complex problems into a series of intermediate, interpretable steps. This paradigm has been naturally extended to Vision-Language Models (VLMs), where early efforts Mitra et al. (2024); Zheng et al. (2023); Lei et al. (2024); Zhang et al. (2023) focused on generating text-only rationales to articulate the model's reasoning process over visual inputs. A pivotal advancement in this domain was the introduction of Interleaved-modal Chain-of-Thought Gao et al. (2025), which pioneered the direct integration of visual patches into the reasoning chain. By pairing textual rationales with corresponding image regions, the interleaved CoT demonstrated a superior ability to ground language in visual evidence, setting a new standard for multimodal reasoning.

However, the efficacy of existing research Ge et al. (2025); Gao et al. (2025) is fundamentally predicated on the reliability of their underlying mechanisms for selecting and integrating visual information. These methods typically rely on a passive, attention-driven strategy: they select the Top-K regions with the highest attention scores and insert them at predefined moments, e.g., the appearance of a newline character, a practice with little theoretical or empirical justification for its timing. This reliance exposes critical vulnerabilities that limit their full potential. Our empirical analysis reveals that high-attention regions are often redundant or, more alarmingly, fail to capture the most crucial visual details, especially when a granularity mismatch exists between the textual

query and the visual evidence. This raises three fundamental questions: (1) What information source can reliably identify truly salient visual regions? (2) How can we accurately select useful regions in a proactive and purposeful manner rather than passively relying on potentially unreliable attention scores? (3) When is the optimal moment to insert visual evidence into the reasoning process?

In this work, we postulate that a more robust multimodal reasoning framework requires a shift from passive attention-following to an active, information-seeking paradigm. We propose AIMCoT: **A**ctive **I**nformation-driven **M**ulti-modal **C**hain-**o**f-**T**hought, a novel framework that directly addresses the limitations of prior work, reframing the selection of visual evidence from a passive, attention-based retrieval task to an active, goal-oriented probing process. Inspired by the principles of information foraging Pirolli & Card (1999); Broadbent (2013), AIMCoT operates on the premise that the most valuable visual rationale is one that maximally reduces the model's uncertainty about the subsequent step in its reasoning chain. Rather than simply asking "Where is the model looking?", we prompt the model to actively ask "Which piece of visual information will be most helpful for me to see *right now*?". This shift is realized through three synergistic components:

1. **Context-enhanced Attention-map Generation (CAG)**, which mitigates the text-vision granularity disparity to improve the reliability of attention map for better identifying salient regions by generating a context-aware description of the image.
2. **Active Visual Probing (AVP)**, which implements a proactive, goal-oriented selection strategy grounded in information theory, selecting a set of visual regions that provide the highest possible information gain for the task at hand.
3. **Dynamic Attention-shifting Trigger (DAT)**, an intelligent triggering mechanism that carefully captures the critical moments when model's cognitive focus shifts significantly from text to vision, and inserts visual information precisely.

Our contributions are as follows:

- We introduce AIMCoT, a novel training-free framework that reframes the construction of multi-modal CoT as an active information-foraging process, moving beyond the limitations of static, passive, and purposeless region selection.
- We propose a system comprised of three complementary methods (CAG, AVP, DAT) that collectively enable VLMs to proactively forage for informative visual evidence and strategically integrate it into their reasoning process at the detected critical moments.
- We present a comprehensive set of empirically-grounded motivations that inspire the designs within our AIMCoT framework. Furthermore, we provide substantial theoretical analysis and design targeted experiments to explore the properties of these designs, covering key aspects such as the deployability of AIMCoT, the interplay between the CAG and AVP modules, and the necessity of incorporating an exploratory candidate pool.
- Through extensive experiments conducted on two backbones (Chameleon-7B and Qwen2-VL-7B) across challenging benchmarks including M3CoT, ScienceQA, and LLaVA-W, we demonstrate that AIMCoT significantly outperforms state-of-the-art baselines and advances the frontier of vision-language reasoning.

## 2 RELATED WORK

The success of CoT prompting in LLMs has naturally extended to VLMs, aiming to make their reasoning processes explicit and interpretable. Early efforts focused on generating text-only rationales. MMCoT Zhang et al. (2023) first generates a rationale from the input and then uses this rationale along with the original multimodal data to infer the final answer. CCoT Mitra et al. (2024) prompts the VLM to generate an intermediate scene graph to structure its understanding. DDCoT Zheng et al. (2023) decomposes complex problems into simpler sub-questions and leverages external models to fill information gaps. Alternatively, SCAFFOLD Lei et al. (2024) overlays a coordinate grid on the image, enabling the VLM to reference spatial regions explicitly in its textual reasoning.

A pivotal advancement is the introduction of interleaved-modal CoT, which integrates visual evidence directly into the reasoning chain. The leading approach, ICoT Gao et al. (2025), selects the top-K regions from the VLM's attention map and inserts them at predefined moments. MRFD Ge et al. (2025) meticulously selects the salient visual regions to foster more reliable reasoning, alleviating

VLM hallucination. However, our analysis reveals that such passive, attention-driven strategies are fundamentally limited by the unreliability of raw attention maps, especially during text-vision granularity mismatches, and their arbitrary insertion points fail to capture the model's dynamic cognitive needs. Our proposed AIMCoT addresses these critical vulnerabilities by shifting the paradigm from passive selection to active information foraging, employing goal-oriented visual probing and dynamic, attention-shift-triggered integration.

## 3 MOTIVATION

### 3.1 IDENTIFYING THE RELIABILITY OF THE ATTENTION MAP

The utilization of the attention map in multimodal learning has been explored by recent research Gao et al. (2025); Ge et al. (2025); Xie et al. (2022); Wang et al. (2023); Liu et al. (2022), inspiring us to also take it as a source from which the salient visual regions are selected. However, considering their strong dependence on attention map, we question its reliability by posing two fundamental questions:

(1) Do all significant regions on the attention map help the VLMs answer questions correctly?
(2) Does the attention map comprehensively capture all visual regions that are instrumental to the VLM's correct prediction?

To acquire a holistic comprehension of these two questions, our empirical analysis is conducted on the popular Visual Question Answering (VQA) benchmark, LLaVA-W Liu et al. (2024), employing ICoT Gao et al. (2025) as a baseline model, which selects the Top-K regions from the attention map to construct a text-vision interleaved CoT. Chameleon-7B Team (2024) serves as the backbone.

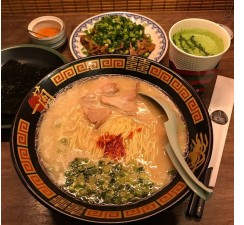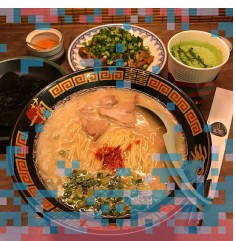

Figure 1: The images of the 22nd question on LLaVA-W benchmark, which is a close-up photo of a meal at ICHIRAN. The left and right figures are respectively the original image and the first three sets of regions selected by the Top-K strategy (red, purple, and blue, respectively). A detailed explanation is shown in Appendix C.

We quantitatively investigate the role of high-attention regions by masking the top-$K_{mask}$ regions identified by ICoT (0-shot). On LLaVA-W, this masking leads to a minor performance drop of just 3.93% (top 10) and 2.44% (top 20), as shown in Table 1. Notably, performance increases when the mask is expanded from the top 10 to the top 20 regions. These results strongly suggest that not all high-attention regions contribute significantly to the model's prediction: some have negligible impact or even introduce detrimental signals.

In response to the second question, we manually inspect the content of high-attention regions to understand their role in VLM's predictions. In a challenging LLaVA-W example (Figure 1), the answer lies in a small area, i.e., the inner rim of a ramen bowl, among rich visual information. The top two most attended image patch sets entirely miss this crucial detail, with only a negligible portion appearing in the third set. This indicates a potential misalignment between high-attention regions and key visual information, particularly when the text-vision granularity disparity is significant.

Table 1: Performance degradation of the baseline model (ICoT, 0-shot) when the Top $K_{mask}$ regions on the attention map are masked.

| $K_{mask}$ | 0 | 1 | 5 | 10 | 20 |
|---|---|---|---|---|---|
| Degradation | 0% | 0.26% | 1.43% | 3.93% | 2.44% |

### 3.2 FORAGING FOR THE MOST INFORMATION TO GUIDE REGION SELECTION

The analysis in Section 3.1 **empirically** reveals that the efficacy of the attention-driven Top-K strategy is notably constrained, particularly when faced with significant text-vision granularity disparities. Furthermore, **fundamentally**, since attention maps merely reflect token correlations, a static approach without an explicit goal like Top-K, which relies solely on attention scores, is inherently suboptimal. This insight directly motivates us to explore a selection method that is more proactive and purposeful.

Existing recognized research Pirolli & Card (1999); Broadbent (2013); Oaksford & Chater (1994); Friston (2010) supports that people, when possible, will maximize their rate of gaining valuable information, as it yields more useful information per unit cost. Inspired by this, we explore providing the VLM with image regions that yield the highest information gain, thereby maximally reducing the model's uncertainty in answering a given question. Intuitively, the model's uncertainty can be quantified by the entropy of its probability distribution over the vocabulary given the current context, while the information gain of an image region is measured based on model's entropy when the region is included in the context. The definitions are meticulously detailed in Section 4.3.

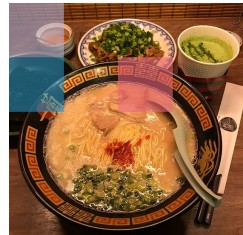

To continue with the example in Section 3.1, Figure 2 illustrates the top three regions ranked by the information gain for VLM. Evidently, in contrast to the regions selected via Top-K shown in Figure 1, the information gain-based selection accurately guides the VLM to first focus on the inner rim of the bowl (red), where the critical information is contained. Although the VLM does not yield the final answer in this region, we note that this indicates a correct line of reasoning, as the ground truth is situated in a highly similar area (a nearby location also on the inner wall of the bowl). Subsequently, the region ranked third (blue) precisely encompasses a large portion of the restaurant's name, which is just the answer to the question. This suggests that even for a challenging case where the text-vision granularity is highly disparate, information gain serves as a better foundation for region selection.

Figure 2: The visualization of regions selected by the information gain-guided strategy.

### 3.3 Inserting Visual Information at Key Moments

Although existing research Gao et al. (2025) has attempted to construct interleaved CoT, they often ignore a reasonable triggering mechanism to capture the critical moment of inserting visual information. For example, ICoT Gao et al. (2025) simply uses a newline character as a trigger signal. This motivates an in-depth investigation into reliable indicators for identifying these key moments for inserting visual content.

To this end, we conduct a comparative case study on the LLaVA-W benchmark, with ROUGE-L as the evaluation metric. We collect all predictions generated by the baseline model (ICoT) and partition them into high- and low-scoring groups based on ROUGE-L scores. Given that the model's attention map has served as an important foundation for existing works Gao et al. (2025); Ge et al. (2025); Xie et al. (2022); Wang et al. (2023); Liu et al. (2022), which is not only readily accessible but also effectively reflects the model's aggregate attention to the textual and visual parts of the input, we are inspired to monitor the attention shifts between these two modalities throughout the prediction process. The experiments are detailed in Appendix G, from which two key observations are revealed: **(1) Correlation analysis:** inserting visual data precisely when the model's attention pivots towards the visual modality is strongly correlated with higher scores and **(2) Group analysis:** this phenomenon further serves as a crucial characteristic that distinguishes high-scoring from low-scoring outputs.

## 4 AIMCoT

In this section, we begin by briefly reviewing the background of multimodal learning. Then, we detail the proposed methods motivated by the following key insights derived from Section 3:

(1) First, the high-scoring attention regions are not always beneficial for question-answering, and crucial visual evidence can be missed particularly in cases of text-vision granularity mismatch.

(2) Second, for a given set of candidate regions, selection based on information gain significantly outperforms the conventional attention-driven Top-K method.

(3) Finally, capturing the critical moments to insert visual information improves the construction of multimodal CoT, and text-to-vision attention shifts serve as an important indicator.

Accordingly, we propose AIMCoT, which encompasses three key methods: (1) **Context-enhanced Attention-map Generation (CAG)**, which generates a fine-grained description for the image to alleviate text-vision disparity. (2) **Active Visual Probing (AVP)**, which proactively and purposefully selects regions that are most helpful for answering the question. (3) **Dynamic Attention-shifting**

**Trigger (DAT)**, which triggers vision insertion into CoT when the model's cognitive focus is significantly shifted from text to vision.

### 4.1 PRELIMINARIES

**Vision-Language Model.** A VLM typically fuses a vision encoder for preprocessing visual input and a generative language model, which jointly enable it to respond in a human-like manner as follows:

$$answer = \text{VLM}(I, x), \tag{1}$$

where $I$ and $x$ are the image and query, respectively.

**Multimodal CoT.** Compared to the direct response shown in Equation 1, multimodal CoT encourages the VLM to output the thought process, called rationales, before outputting the final answer. Existing work Gao et al. (2025) has been devoted to generalizing rationales from pure text to text-vision interleaved form.

### 4.2 CONTEXT-ENHANCED ATTENTION-MAP GENERATION (CAG)

In this component, the VLM is prompted to carefully generate an explanatory description of the given image within the question before the process of VQA, with the explicit goal of helping a potential respondent correctly answer the question. In this process, the VLM acts as a facilitator who interprets the image in the context of the given question to guide the respondent's thought process. Formally, it is expressed as follows:

$$\mathcal{D}_{CAG} = \text{VLM}(I, x, \mathcal{P}_{CAG}), \tag{2}$$

where VLM, $I, x$ are the used VLM, the given image and question, respectively. $\mathcal{P}_{CAG}$ is the prompt provided for the model to generate the description. Then, to compensate for the sparsity of textual information within the context, the generated description $\mathcal{D}_{CAG}$ is concatenated to the question $x$ as follows:

$$x' = \text{concat}(x, \mathcal{D}_{CAG}). \tag{3}$$

By enhancing the context to compensate for the sparsity of textual information, the disparity in text-vision granularity is effectively mitigated, unlocking the potential for the attention map to serve as a more reliable indicator of task-relevant regions. We provide an example in Appendix H.1 that details the template of $\mathcal{P}_{CAG}$, the entire process of CAG, and how the final attention map $A'$ is generated, which is used in the next stage.

### 4.3 ACTIVE VISUAL PROBING (AVP)

Built upon the theoretical foundation of information gain, AVP is designed to select the crucial regions from a set of candidate visual regions. Although AVP consists of three steps elaborated upon as follows, in terms of complexity, our provided analysis and empirical results in Appendices K and M suggest that the introduction of AVP still enables AIMCoT to strike a good balance between deployability and superior performance. A visualization of AVP is shown in Figure 3.

**Diversified Set of Candidate Regions Construction.** Based on the first two insights summarized in Section 4, an inference can be obtained: relying solely on attention maps as the source may prevent the optimal region selection. Motivated by this, we propose to diversify the source of the candidate regions, thereby reducing the model's dependency on attention maps alone. Specifically, we not only construct an attention-driven candidate set, $C_{attn}$, by selecting $N$ regions with the highest attention scores ($N \in \mathbb{R}$), but also generate an exploratory candidate set, $C_{exp}$, by sampling $M$ grid regions uniformly at random from the input image ($M \in \mathbb{R}$). Our empirical analysis in Appendix J shows that the incorporation of exploratory set $C_{exp}$ provides substantial salient visual regions for the VLM. Furthermore, we study the impact of introducing $C_{exp}$ on the performance of AIMCoT, and further compare random sampling, selective search Uijlings et al. (2013), and FastSAM Zhao et al. (2023) as methods for constructing $C_{exp}$ in Appendix N.

Formally, the process can be expressed as follows:

$$C_{attn} = \{R_1, R_2, \cdots, R_N\}, \quad \text{s.t. } R_i = \text{Top-}i \text{ Region from the attention map } A', \tag{4}$$

$$C_{exp} = \{R_{N+1}, R_{N+2}, \cdots, R_{N+M}\}, \quad C = C_{attn} \cup C_{exp}, \tag{5}$$

Figure 3: An overview of our AVP module, which iteratively selects $K$ most informative regions from a diversified candidate set $C$ to build an interleaved CoT that fosters vision-language reasoning.

where $C_{attn}, C_{exp}$, and $C$ are the attention-driven, exploratory, and total candidate sets, respectively. Subsequently, the goal is to select the most informative $K$ regions from $C$ ($K < |C| = N + M$).

**Quantification of Information Gain.** An ideal region of input image is the one that is most informative and critical for answering the question, i.e., the one that minimizes the model's predictive uncertainty. Based on this idea, the region selection problem is transformed into a sequential information gain maximization task. To formalize this objective, we propose the concept of **basic uncertainty** $U_B$ defined as follows:

$$U_B = H(Y|I, x, y_{<t}) = -\sum_{y \in V} P(y|I, x, y_{<t}) \log_2 P(y|I, x, y_{<t}), \tag{6}$$

where $I, x$ and $y_{<t}$ are the input image, question and the predicted tokens, respectively. $P(y|I, x, y_{<t})$ is model's probability distribution across the vocabulary $V$ in current context without the introduction of any region $R_i \in C$. It is seen that $U_B$ takes the form of entropy, thereby inherently capturing the model's uncertainty when predicting the next token $t$ without incorporating any $R_i \in C$.

Similarly, the **conditional uncertainty** $U_{C,i}$, which illustrates the model's uncertainty when an arbitrary region $R_i \in C$ is included into the context, is proposed and defined as follows:

$$U_{C,i} = H(Y|I, x, y_{<t}, R_i) = -\sum_{y \in V} P(y|I, x, y_{<t}, R_i) \log_2 P(y|I, x, y_{<t}, R_i), \tag{7}$$

where $P(y|I, x, y_{<t}, R_i)$ is the model's probability distribution across the vocabulary $V$ in the context combined with the region $R_i$. Eventually, the aforementioned conceptual formulation logically motivates the definition of our final key metric: **the information gain of region** $R_i$, which is formally defined as follows:

$$IG(\{R_i\}) = U_B - U_{C,i}, \quad i = 1, 2, \cdots, N + M. \tag{8}$$

Intuitively, $IG(\{R_i\})$ quantitatively characterizes how incorporating region $R_i$ reduces the model's uncertainty for the subsequent token prediction.

**Optimal Region Selection.** To achieve the target of region selection, we formalize the problem as follows:

$$\text{Maximize } F(S) = IG(S), \quad \text{s.t. } S \subset C, |S| = K, \tag{9}$$

where $S$ is the optimal set of selected regions and $K$ is the size of $S$ serving as a hyper-parameter. Motivated by two crucial insights, we propose a greedy algorithm detailed in Algorithm 1 to solve this problem: (1) According to recognized works Bian et al. (2017); Sener & Savarese (2018); Kim et al. (2016); Krause et al. (2008); Das & Kempe (2011), the greedy algorithm is a well-established and widely-used method for maximizing functions, especially for maximizing those that exhibit a tendency towards submodularity, **even if they are not submodular theoretically**; (2) Our experiments in Appendix I suggest that the information gain function $F$ empirically exhibits significant approximate submodularity.

Specifically, the selection is an iterative process consisting of $K$ steps. At each step, AVP proactively selects the most informative and not-yet-chosen region from the candidate set $C$ with the explicit goal of minimizing the uncertainty of the model's answer to the question. The chosen region is then added to an intermediate set $R^*$. Notably, a key merit of AVP is its ability to bypass regions that, despite high attention scores, exhibit strong informational overlap with regions that have been selected. After $K$ rounds, the collection of all selected regions in $R^*$ constitutes the final optimal selection $S$. Substantial results in Appendices K.2 and M suggest that our framework's average inference time is no more than 1.36 times that of an efficient baseline ICoT, and the framework scales efficiently with larger values of $K$ and $N_C = |C|$, respectively.

## 4.4 Dynamic Attention-shifting Trigger (DAT)

As motivated by the key observation in Section 3.3, it is vital to appropriately time the insertion of visual information when constructing a multimodal CoT, and furthermore, the shift of attention from the textual to the visual context serves as a crucial indicator. Motivated by these insights, we propose the DAT mechanism, which systematically evaluates the model's attention scores on the visual context at every token $t$ generation step formulated as follows:

$$A_{visual}(t) = \sum_{i \in \text{indices of } C_{visual}} \bar{a}_{t,i}, \tag{10}$$

where $C_{visual}$ is the visual information within the context; $\bar{a}_{t,i}$ is the average attention score of token $t$ towards the visual token with index $i$ across the last $N_L$ VLM layers. Drawing inspiration from the NLP community Jawahar et al. (2019); Tenney et al. (2019); Vig & Belinkov (2019), we restrict our focus to the model's final layers, as they are responsible for capturing high-level semantic information, including semantic roles and coreference relations, where the shifting signal is presumed to be more reliable. In our implementation, we use the last 3 layers by default ($N_L = 3$). Then, the shift of attention is formalized as follows:

$$\Delta A_{visual}(t) = A_{visual}(t) - A_{visual}(t - 1), \tag{11}$$

which quantifies the model's attention shift towards the visual context between generating the current token and the preceding one. Eventually, a hyper-parameter $\delta \in \mathbb{R}$ is employed to delineate the point at which the attention shift $\Delta A_{visual}(t)$ is substantial enough to activate the AVP to insert essential visual information in the multimodal CoT. We detail a sensitivity analysis of the threshold $\delta$ with an emphasis on its impact on the frequency of triggering AVP and the performance of AIMCoT in Appendix L.

## 5 Experiments

### 5.1 Benchmarks and Baselines

In this study, we evaluate AIMCoT on three popular and challenging VQA benchmarks, including **M3CoT** Chen et al. (2024), **ScienceQA** Saikh et al. (2022), and **LLaVA-Bench In-the-Wild** (LLaVA-W) Liu et al. (2024). We provide detailed introductions in Appendix D.

To evaluate the performance of AIMCoT, we introduce the vanilla VLM w/o CoT (No-CoT) and a range of state-of-the-art methods as baseline models, including DDCoT Zheng et al. (2023), MMCoT Zhang et al. (2023), CCoT Mitra et al. (2024), and SCAFFOLD Lei et al. (2024), which generate text-only rationales. Furthermore, ICoT Gao et al. (2025), which constructs interleaved-modal CoT, is considered as well. The detailed introduction to them is listed in Appendix E. In presenting the results, we directly cite the performance reported in existing works where applicable.

### 5.2 Implementation Details

We implement AIMCoT and the baselines on Chameleon-7B Team (2024) and Qwen2-VL-7B-Instruct Wang et al. (2024) in two settings (both 0- and 1-shot), which aligns with the recent leading research Gao et al. (2025). The experiments are conducted on A6000 GPUs. The hyper-parameter settings for AIMCoT and the reproducibility statement are meticulously listed in Appendix H.2 and Section 7, respectively.

Table 2: Performance comparison results on three widely-used benchmarks. The best performances are shown in bold. The metric for experiments on M3CoT and ScienceQA is Accuracy (ACC.), while on LLaVA-W, the metric ROUGE-L is adopted.

| Backbone | Method | M3CoT (ACC.) | | ScienceQA (ACC.) | | LLaVA-W (ROUGE-L) | |
|---|---|---|---|---|---|---|---|
| | | 0-shot | 1-shot | 0-shot | 1-shot | 0-shot | 1-shot |
| Chameleon-7B | No-CoT | 29.1 | 28.4 | 47.7 | 48.5 | 13.1 | 23.9 |
| | DDCoT | 28.6 | 29.8 | 49.8 | 49.2 | 20.2 | 23.1 |
| | MMCoT | 28.5 | 30.6 | 49.0 | 50.7 | 20.4 | 20.6 |
| | CCoT | 29.4 | 31.4 | 50.2 | 51.3 | 22.1 | 24.5 |
| | SCAFFOLD | 29.6 | 31.1 | 48.5 | 47.5 | 21.7 | 24.7 |
| | ICoT | 29.8 | 32.3 | 51.0 | 53.4 | 25.2 | 27.6 |
| | AIMCoT (Ours) | **31.4** | **32.8** | **53.1** | **54.5** | **29.8** | **32.0** |
| | Improvement | 5.50% | 1.47% | 4.08% | 2.04% | 18.25% | 15.94% |
| Qwen2-VL-7B | No-CoT | 43.6 | 45.4 | 56.3 | 64.4 | 32.7 | 33.5 |
| | MMCoT | 40.1 | 42.5 | 51.3 | 58.3 | 30.7 | 31.4 |
| | CCoT | 43.3 | 44.1 | 56.4 | 63.8 | 29.4 | 33.9 |
| | DDCoT | 42.6 | 45.7 | 55.2 | 64.9 | 31.2 | 32.8 |
| | SCAFFOLD | 41.7 | 44.9 | 53.7 | 62.5 | 31.8 | 33.1 |
| | ICoT | 44.1 | 46.0 | 56.8 | 65.4 | 34.2 | 35.7 |
| | AIMCoT (Ours) | **44.7** | **46.6** | **57.4** | **66.3** | **36.3** | **37.3** |
| | Improvement | 1.4% | 1.3% | 1.1% | 1.3% | 6.2% | 4.5% |

## 5.3 Performance Comparison

We evaluate the performance of AIMCoT against the state-of-the-art (SOTA) methods. The results shown in Table 2 clearly demonstrate the superiority of our proposed AIMCoT as it significantly outperforms all the baseline models under both 0- and 1-shot settings across all the datasets.

Specifically, AIMCoT surpasses all baselines that generate text-only rationales, confirming the efficacy of integrating salient visual information directly into CoT. When compared to ICoT, which also produces interleaved text-vision CoT, AIMCoT's superior performance underscores the importance of our three key contributions: (1) a more reliable attention map as a foundation, (2) a proactive, goal-oriented mechanism for image region selection, and (3) an intelligent trigger for inserting visual information at critical moments.

Crucially, AIMCoT's advantage is most pronounced on the open-ended LLaVA-W benchmark and in the 0-shot setting, which better simulate complex, real-world scenarios where the model must rely solely on its internal knowledge and reasoning. By emulating what can be seen as a more human-like cognitive process, AIMCoT unlocks the VLM's foundational reasoning capabilities, enabling robust performance in novel and challenging situations.

## 5.4 Ablation Study

In this section, we conduct a series of ablation studies to verify the efficacy of each component within AIMCoT. The details of settings are as follows:

- In **w/o CAG**, the VLM is directly prompted with the **raw** question $x$ and the paired image $I$;
- In **w/o AVP**, the AVP is replaced by the attention-driven Top-K strategy by following existing works Gao et al. (2025); Ge et al. (2025), which selects the regions with Top-K attention scores on the model's attention map;
- In **w/o DAT**, following existing research Gao et al. (2025), the insertion of visual information in CoT is triggered when the model outputs the signal token, which is a line break by default.

Ablation results in Table 3 validate the contributions of our core components. First, CAG provides essential context enhancement, proving crucial for generating high-quality CoT, particularly when text queries are sparse (M3CoT, LLaVA-W). Second, the consistent, significant performance drop

when replacing AVP with a Top-K baseline underscores the substantial superiority of our proactive, information-oriented method. Furthermore, the removal of DAT also results in a considerable performance decline, highlighting the critical importance of the timing of image insertion.

The performance gains from AIMCoT are most pronounced on LLaVA-W, a challenging benchmark requiring open-ended generation. This large improvement starkly demonstrates our model's

Table 3: Ablation study of AIMCoT conducted on Chameleon-7B under 0-shot setting.

| Dataset | AIMCoT | w/o CAG | w/o AVP | w/o DAT |
|---|---|---|---|---|
| M3CoT (ACC.) | 31.4 | 30.5 (-0.9) | 30.6 (-0.8) | 30.8 (-0.6) |
| ScienceQA (ACC.) | 53.1 | 52.8 (-0.3) | 52.3 (-0.8) | 52.7 (-0.4) |
| LLaVA-W (ROUGE-L) | 29.8 | 26.8 (-3.0) | 26.2 (-3.6) | 27.3 (-2.5) |

advanced capability to comprehend intricate multimodal information and tackle demanding, unconstrained tasks.

### 5.5 In-depth Analysis: the Interplay between CAG and AVP

Our proposed components collaborate organically to foster the construction of multimodal CoT: as a preceding module, CAG enriches the context to benefit the construc-

Table 4: Ablation study of the baseline model (BM) on Chameleon-7B under 0-shot setting.

| Dataset | BM | BM w/ CAG | BM w/ AVP | BM w/ CAG, AVP |
|---|---|---|---|---|
| M3CoT (ACC.) | 29.8 | 30.3 (+0.5) | 30.2 (+0.4) | 30.8 (+1.0) |
| ScienceQA (ACC.) | 51.0 | 52.0 (+1.0) | 51.9 (+0.9) | 52.7 (+1.7) |
| LLaVA-W (ROUGE-L) | 25.2 | 25.8 (+0.6) | 26.4 (+1.2) | 27.3 (+2.1) |

tion of the candidate set; subsequently, AVP triggered by DAT proactively selects the most salient regions from the candidate set.

In this section, we investigate the interaction between CAG and AVP via an ablation study. Starting from the baseline ICoT model (BM), i.e., AIMCoT stripped of all proposed modules, we sequentially add CAG and then AVP. Table 4 presents the results, from which we derive the following insights:

- **The compatibility between CAG and AVP:** both CAG and AVP individually provide significant gains, but their combination synergistically improves the construction of the interleaved CoT.
- **The consistent superiority of AVP over Top-K selection:** AVP consistently and significantly outperforms the standard Top-K selection method for choosing image regions, both with and without the presence of the CAG module.
- **The interplay between CAG and AVP:** The average performance improvement of AVP over Top-K selection increases from 2.62% to 2.94% when CAG is introduced. This suggests CAG enhances the source attention map, providing a more reliable set of candidate regions and thereby unlocking AVP's full potential to select the most salient visual evidence.

## 6 Conclusion

In this paper, we propose AIMCoT, a novel framework that reframes the construction of interleaved-modal CoT as an active, information-foraging process, addressing the limitations in existing methods, which often rely on passive, heuristic-driven mechanisms for selecting and inserting visual information at suboptimal moments. Our extensive experiments on three popular and challenging benchmarks demonstrate that AIMCoT significantly outperforms state-of-the-art methods in both 0- and 1-shot settings (up to 18%). By dynamically structuring its reasoning and actively seeking the most informative visual cues, AIMCoT achieves a more proactive, goal-oriented, and human-like approach to vision-language reasoning.

Despite the strong performance, AIMCoT presents avenues for future exploration. The AVP module, while highly effective and optimized, introduces a slight computational overhead compared to simpler attention-based selection. Future work could explore lightweight, learnable policies for region selection to further enhance its deployability. We also plan to extend our evaluation to a broader range of VLM architectures and more complex, long-form reasoning tasks to further probe the generalizability and limits of our active information-seeking paradigm.

## 7 REPRODUCIBILITY STATEMENT

To ensure the reproducibility of our work, we provide the complete source code for our AIMCoT framework and all experiments in the supplementary materials, accessible via an anonymized link: https://anonymous.4open.science/r/AIMCoT. The architectural details and theoretical underpinnings of our proposed components, including Context-enhanced Attention-map Generation (CAG), Active Visual Probing (AVP), and Dynamic Attention-shifting Trigger (DAT), are thoroughly described in Section 4 of the main paper. The specific greedy algorithm employed by AVP is detailed in Algorithm 1 (Appendix F). All datasets used in our evaluation are publicly available benchmarks, as detailed in Section 5.1 and Appendix D. We provide a comprehensive list of all hyper-parameter settings used to achieve the reported results for each benchmark in Appendix H.2 (Table 5). Furthermore, extensive ablation studies (Section 5.4), and in-depth analyses (Section 5.5, Appendices G,I, J, L, and M), are provided to allow for a complete replication of our findings.

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

## A    Appendix

## B    LLM Usage Statement

In the preparation of this manuscript, we utilized a large language model (LLM) as a general-purpose writing assistance tool. The primary uses of the LLM were twofold:

- Grammatical Correction: The LLM was employed to proofread the manuscript for grammatical errors, spelling mistakes, and awkward phrasing. This helped improve the overall clarity and readability of our paper.
- Text Condensation: To adhere to the page limits of the conference, we used the LLM to help condense and rephrase certain paragraphs and sentences.

All suggestions provided by the LLM were carefully reviewed, critically evaluated, and manually edited by the authors to ensure that the scientific integrity and intended meaning of our work were preserved. Notably, the LLM was not used for core research activities, including the ideation of the AIMCoT framework, the design of experiments, the analysis of results, or the generation of the primary scientific claims. The final intellectual content and all contributions presented in this paper are entirely our own.

## C    Explanation for Figure 1

Figure 1 shows the images of the 22nd question on LLaVA-W benchmark. The query for this question is: *"What's the name of the restaurant serving these dishes?"* and the image is a close-up photo of a meal at ICHIRAN. The left figure is the original image, and the right figure visualizes the regions selected by the Top-K strategy. The $num\_selected\_patches$ regions with the highest, second-highest, and third-highest scores are designated as the first, second, and third sets, colored red, purple, and blue, respectively. The first set is utilized by the baseline model according to its default design ($num\_selected\_patches = 72$).

## D    Introduction to the Benchmarks

**M3CoT** Chen et al. (2024) is a novel multimodal CoT benchmark, which introduces complex, multi-step problems across science, mathematics, and commonsense domains, comprising 11,459

samples in total. M3CoT is characterized by succinct textual queries (<15 tokens on average) paired with intricate problems. This inherent text-vision imbalance makes it an ideal platform to validate the efficacy of our proposed CAG in mitigating this issue and the superiority of AVP in proactively selecting the salient visual regions.

**ScienceQA** Saikh et al. (2022) is a popular benchmark for multiple-choice question answering with explanations on scholarly articles, comprising over 100,000 context-question-answer triples to address data scarcity in scientific machine reading comprehension.

**LLaVA-Bench In-the-Wild** (LLaVA-W) Liu et al. (2024) is a challenging open-ended benchmark designed to evaluate the real-world capabilities of VLMs by mimicking the unpredictability of real-world scenarios. The answers generated by GPT-4v Achiam et al. (2023) serve as the labels. LLaVA-W is exceptionally well-suited for evaluating the capability of our proposed framework to address complex, open-ended problems by generating a multimodal CoT, attending to salient regions within the image, and meticulously parsing the query.

## E  INTRODUCTION TO THE BASELINE MODELS

**No-CoT** prompts the VLM to answer questions directly based on the input query and image. In the 1-shot setting, an example containing the query, image, and corresponding answer is attached.

**DDCoT** Zheng et al. (2023) deconstructs a multimodal problem into reasoning and recognition sub-questions, uses negative-space prompting to identify and fill visual information gaps with external models, and then integrates all information for a final joint reasoning step to generate rationales.

**MMCoT** Zhang et al. (2023) first generates a rationale from fused language and vision inputs, and then uses this rationale along with the original multimodal data to infer the final answer.

**CCoT** Mitra et al. (2024) first prompts the VLM to generate a scene graph from an image and then uses it as an intermediate reasoning step to produce the final response.

**SCAFFOLD** Lei et al. (2024) promotes vision-language coordination in the VLM by overlaying a dot matrix with coordinates onto an image, which then serves as a visual anchor that can be explicitly referenced in the textual prompt.

**ICoT** Gao et al. (2025) leverages the attention maps of the VLM to select relevant patches from the input image and insert them into the reasoning process, thereby generating sequential steps of paired visual and textual rationales.

## F  GREEDY ALGORITHM WITHIN AVP MODULE

The complete process of the greedy algorithm within AVP is shown in Algorithm 1.

---

**Algorithm 1:** Greedy Algorithm for Optimal Region Selection

**Input:** total candidate set $C$, size of optimal selection $K$
**Output:** optimal selection $S$

1  $R^* \leftarrow \varnothing$
2  **for** $k \leftarrow 1, 2, \cdots, K$ **do**
3      $U_B \leftarrow H(Y|I, x, y_{<t}, R^*)$
4      **for** $i \leftarrow 1, 2, \cdots, N + M$ **do**
5         $U_{C,i} \leftarrow H(Y|I, x, y_{<t}, R^* \cup \{R_i\})$
6         $IG(\{R_i\}) \leftarrow U_B - U_{C,i}$
7      $R_{next} \leftarrow \text{argmax}_{R_i \in C \setminus R^*}\{IG(\{R_i\})\}$
8      $R^* \leftarrow R^* \cup \{R_{next}\}$
9  $S \leftarrow R^*$
10 **return** $S$

---

## G    DETAILED ANALYSIS OF KEY MOMENTS TO INSERT VISUAL INFORMATION

**Experimental Setup.** We take ICoT Gao et al. (2025) as a baseline model, which is required to answer all questions from the LLaVA-W benchmark in a 0-shot setting, with ROUGE-L used as the evaluation metric. The hyper-parameters follow the default settings of the open-source implementation for ICoT, and all experiments are conducted with the Chameleon-7B backbone.

**Formal Definition of Attention Shifts.** To analyze attention shifts, we examine the averaged attention maps across all attention heads in the last three layers of the VLM during the prediction of each token $t$. The model's total attention scores allocated to the visual and text components of the input are respectively measured as follows:

$$A_{visual}(t) = \sum_{i \in \text{indices of } C_{visual}} \bar{a}_{t,i}, \qquad A_{text}(t) = \sum_{j \in \text{indices of } C_{text}} \bar{a}_{t,j}, \qquad (12)$$

where $C_{visual}, C_{text}$ are the visual and text information within the context, respectively. Then, the shift in attention from the textual to the visual modality while generating token $t$ is defined as follows:

$$\delta_t = A_{visual}(t) - A_{visual}(t-1). \qquad (13)$$

$\Delta_k = [\delta_1, \delta_2, \cdots, \delta_{|\Delta_k|}]$ encompasses the model's attention shifts for each token when answering the arbitrary $k$-th question, where $|\Delta_k|$ is the number of tokens for answering the $k$-th question.

**Formal Definition of Scores.** for the predictions generated by the baseline model, the ROUGE-L scores are given by $List\_R = [R_1, R_2, \cdots, R_{|List\_R|}]$, where $R_k$ is the score for the model's response to the $k$-th question, and $|List\_R|$ is the number of questions within the benchmark.

Based on these concepts, we design a two-part experiment:

**Experiment 1: Correlation Analysis.** We investigate the relationship between the proportion of visual insertions under significant attention shifts and the score of the corresponding generated prediction.

First, to identify whether a visual insertion is conducted during a significant attention shift, we define a high attention growth threshold, $\delta_k^{(h)}$ for the $k$-th response ($\delta_k^{(h)}$ is set to the 80% upper quantile of $\Delta_k$ by default). An insertion is considered to have been conducted under a significant shift and referred to as a *synchronized insertion* if and only if its corresponding attention shift value exceeds the threshold $\delta_k^{(h)}$.

Next, since the model can conduct multiple insertions per response for a question, we calculate $P_k$, the proportion of synchronized insertions out of the total number of insertions for the $k$-th question.

Finally, since the proportions of synchronized insertions $[P_1, P_2, \cdots, P_{|List\_R|}]$ and the ROUGE-L scores for all the questions $[R_1, R_2, \cdots, R_{|List\_R|}]$ are obtained, the Pearson Correlation coefficient can be computed. Specifically, the Pearson Correlation is 0.2166 with a p-value of 0.048, which suggests that the proportions of the synchronized insertions and the corresponding score are significantly positively related to each other.

**Experiment 2: Group Analysis.** We investigate the relationship between the proportion of synchronized insertions and the quality of the model's response.

To group the generated predictions according to response quality, we establish high- and low-scoring groups. All predictions are ranked in descending order by their ROUGE-L scores. The top 30% form $G_h$, the high-scoring group (high-quality responses), and $G_l$, the bottom 30% form the low-scoring group (low-quality responses).

Then, we calculate the mean proportion of synchronized insertions for groups $G_h, G_l$, which are denoted as $\bar{P}_h, \bar{P}_l$, respectively.

Finally, the means of the two groups $\bar{P}_h, \bar{P}_l$ are compared, and a T-test is performed to assess the statistical significance of the difference. Specifically, we find that $\bar{P}_h = 0.8889, \bar{P}_l = 0.5000$, which suggests that in the high-scoring group, approximately 89% of insertions are the synchronized insertions with significant attention shift from textual input to visual information; in contrast, in the low-scoring group, only about half of the insertions are synchronized insertions. Besides, the P-value of T-test is as low as 0.0019, which demonstrates that the result is highly statistically significant.

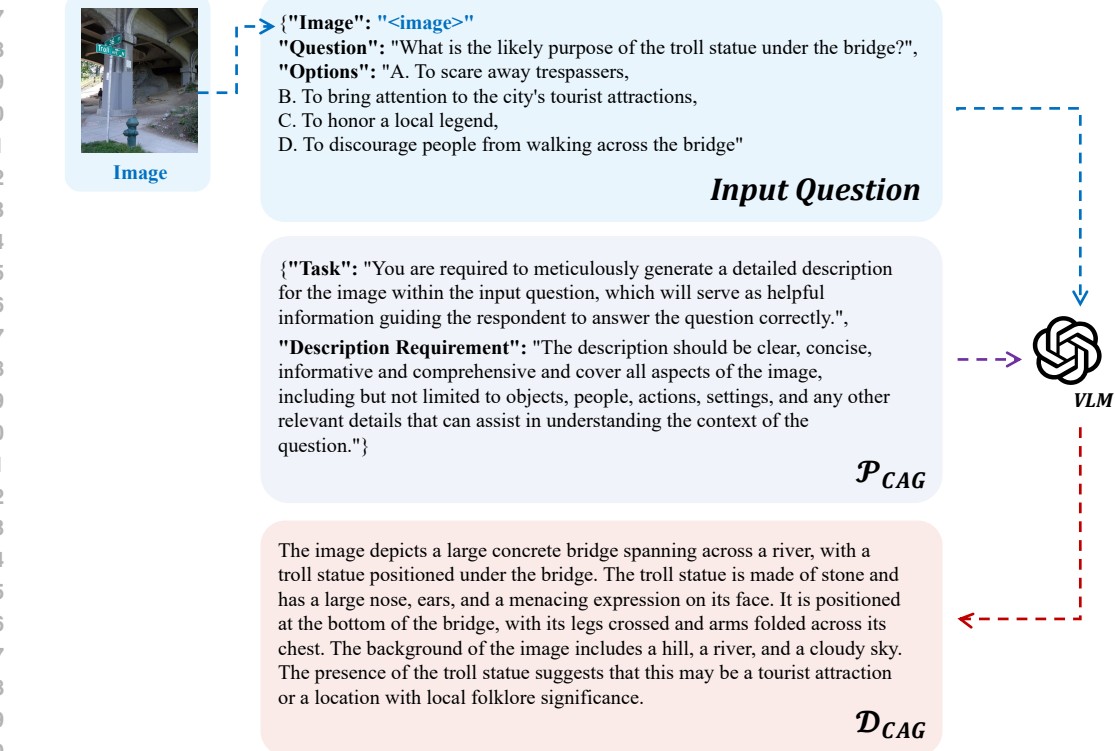

Figure 4: An illustration of the entire process of context enhancement by the CAG module, using problem physical-commonsense-1398 from the M3CoT benchmark as an example. This showcases both the template and usage of $\mathcal{P}_{CAG}$.

## H MODEL SETTINGS

### H.1 TEMPLATE OF $\mathcal{P}_{CAG}$

Figure 4 provides an intuitive example showing the template of $\mathcal{P}_{CAG}$ and how it is used to prompt the VLM to carefully generate a guiding description for the input image. In particular, for multiple-choice questions, such as those in M3CoT and ScienceQA, we prepend the following brief explanation to $\mathcal{P}_{CAG}$ to aid the VLM in better understanding its designated task: "*This is a multiple-choice question. The question is based on the image provided.*"

Then, the cross-attention weight matrix based on the enhanced context $x'$ and $I$ can be obtained as follows:

$$A' = \text{softmax}\left(\frac{(H_T W^Q)(H_V W^K)^T}{\sqrt{d_K}}\right), \tag{14}$$

where $H_T \in \mathbb{R}^{n_T \times d}, H_V \in \mathbb{R}^{n_V \times d}$ are the hidden states of the textual and visual input, respectively. $W^Q, W^K$ are the weight matrices of the linear transformation layers for query and key, respectively.

### H.2 SETTING OF HYPER-PARAMETERS

The hyper-parameter settings are presented in Table 5. For easier understanding, we provide further explanation of the hyper-parameters $s_r$ and $s_g$ as follows: in the AVP module, the input image is first divided into $s_g \times s_g$ grids according to the set "Grid size" $s_g$. Then, each region finally selected by AVP consists of $s_r \times s_r$ grids.

Table 5: Hyper-parameter settings across three datasets.

| Parameter | M3CoT | ScienceQA | LLaVA-W |
|---|---|---|---|
| $N_C$ | 8 | 8 | 6 |
| $K$ | 3 | 3 | 3 |
| $N$ | 4 | 4 | 2 |
| $M$ | 4 | 4 | 1 |
| Region size for AVP $s_r$ (grid) | 1 | 1 | 1 |
| Grid size for AVP $s_g$ | 4 | 4 | 4 |
| $\delta$ | 0.5 | 0.2 | 0.2 |
| Temperature | 0.7 | 0.7 | 0.7 |
| Do sample | True | True | True |
| Top_p | 0.9 | 0.9 | 0.9 |
| Repetition_penalty | 1.2 | 1.2 | 1.2 |
| Min_new_tokens | 32 | 32 | 32 |
| Max_new_tokens | 512 | 1024 | 1024 |

## I    EMPIRICAL ANALYSIS OF THE APPROXIMATE SUBMODULARITY OF FUNCTION $F$

To offer a more comprehensive insight into the motivation for employing a greedy algorithm, this section provides a thorough analysis. We emphasize that for functions that are not theoretically submodular, a greedy approach remains one of the conventional methods for addressing their maximization, as established in recognized works Bian et al. (2017); Sener & Savarese (2018); Kim et al. (2016); Krause et al. (2008); Das & Kempe (2011). In this part, we conduct meticulously designed experiments to investigate the extent to which the information gain function $F$ approximates submodularity. The experimental results reveal that $F$ empirically exhibits significant submodular characteristics. This finding motivates us to follow established works Bian et al. (2017); Sener & Savarese (2018); Kim et al. (2016); Krause et al. (2008); Das & Kempe (2011) to propose a greedy algorithm to solve the maximization problem for $F$. The analysis is detailed as follows:

Firstly, we would like to introduce the definition of a submodular function. According to existing research Nemhauser et al. (1978), a function $f$ is a submodular function if it satisfies

$$f(A \cup \{R_i\}) - f(A) \geq f(B \cup \{R_i\}) - f(B) \tag{15}$$

for any sets $A \subseteq B \subset C$ and any element that satisfies $R_i \in C \backslash B$. In our scenario, the Inequality 15 is written as

$$F(A \cup \{R_i\}) - F(A) \geq F(B \cup \{R_i\}) - F(B) \tag{16}$$

for any $A \subset B \subset C$ and any $R_i \in C \backslash B$, which means that the information gain from incorporating a visual region exhibits a diminishing returns property.

To demonstrate this empirically, we design the experiment detailed as follows, aiming to show that for two sets of regions of different sizes, $S_{small} \subset S_{large} \subset C$, the information gain from incorporating a given visual region $R_{test} \in C \setminus S_{large}$ into the context of a VLM is greater when $R_{test}$ is added to $S_{small}$ than when it is added to $S_{large}$, ceteris paribus.

**Experimental Setup.** In our experimental design, each time the AVP process is triggered to select salient regions, we first execute it to select $K_{small}$ regions from the total candidate pool $C$ to form the set $S_{small}$. Subsequently, building upon $S_{small}$, we select an additional $K_{large} - K_{small}$ regions to construct the set $S_{large}$, where $K_{small}$ and $K_{large}$ are the respective set sizes. This construction inherently ensures that $S_{small} \subset S_{large}$.

Next, to compute the information gain contributed by a given region, we randomly sample a region $R_{test}$ from $C \setminus S_{large}$. We then calculate the VLM's information content, which are denoted as $U_s, U_s^*, U_l$, and $U_l^*$, when the context incorporates (1) $S_{small}$, (2) $S_{small} \cup \{R_{test}\}$, (3) $S_{large}$, and (4) $S_{large} \cup \{R_{test}\}$, respectively. We expect to observe in the majority of cases that:

$$U_s^* - U_s \geq U_l^* - U_l. \tag{17}$$

We conduct experiments on the M3CoT and LLaVA-W benchmarks, setting $K_{small} \in \{2, 3, 4, 5\}$ and $K_{large} = K_{small} + 1$ for simplicity. In terms of evaluation, we record the proportion of instances for which the inequality $U_s^* - U_s \geq U_l^* - U_l$ holds, and further introduce a Binomial Test to rigorously examine the significance of the results.

**Experimental Results.** The experimental results are presented in Table 6. As we can see, the Inequality 15 holds in most instances across all settings and datasets. Furthermore, to confirm the significance of the obtained results, we introduce the Binomial Test, an exact statistical procedure for assessing the extent to which experimental outcomes with a binary structure are attributable to chance alone. The p-values, presented in Table 6, are all substantially below the 0.05 significance level. This demonstrates that the information gain function $F$ behaves in a manner that is empirically near-submodular, which motivates us to follow existing research Bian et al. (2017); Sener & Savarese (2018); Kim et al. (2016); Krause et al. (2008); Das & Kempe (2011) where greedy algorithms are proposed to solve the problem of maximizing approximately submodular functions.

Table 6: Proportions of instances on M3CoT and LLaVA-W benchmarks for which the approximate submodularity of information gain function $F$ is manifested. The backbone is Chameleon-7B and the model is our proposed AIMCoT. $K_{large}$ is set to $K_{small} + 1$ for simplicity. The significance levels of these results are listed below them.

| $K_{small}$ | 2 | 3 | 4 | 5 |
|---|---|---|---|---|
| M3CoT (n=2318) | 72.00% | 62.99% | 67.04% | 61.09% |
| P-value | <1e-6 | <1e-6 | <1e-6 | <1e-6 |
| LLaVA-W (n=60) | 61.67% | 68.33% | 61.67% | 63.33% |
| P-value | 0.0462 | 0.0031 | 0.0462 | 0.0249 |

## J ANALYSIS OF THE SELECTED REGIONS' SOURCE

In this section, we examine the distribution of sources for the visual regions selected by the AVP module of AIMCoT. These regions are drawn from two sets, $C_{attn}$ and $C_{exp}$, with their respective selection proportions denoted as $P_{attn}$ and $P_{exp}$. Intuitively, $P_{exp}$ reflects the significance of incorporating the exploratory set $C_{exp}$ to construct a better multimodal CoT. A larger value of $P_{exp}$ indicates that the exploratory set $C_{exp}$ makes a greater contribution by providing informative salient regions to AIMCoT, and vice versa.

**Experimental Setup** The experiments are conducted on the M3CoT and LLaVA-W benchmarks. Our proposed AIMCoT is implemented with the Chameleon-7B backbone under a default 0-shot setting. To ensure the reliability of the results, we repeat each experiment three times on both benchmarks.

**Results and Analysis** As presented in Table 7, although the value of $P_{exp}$ fluctuates across different experimental runs on the same benchmark, it remains consistently around 20% on M3CoT and 30% on LLaVA-W. This indicates that the influence of stochastic factors on the source distribution of the selected regions is limited, which validates our rationale of using this metric as a reflection of the relative importance of $C_{attn}$ and $C_{exp}$. Furthermore, we observe that $P_{exp}$ is significantly greater than zero. This demonstrates that the exploratory set $C_{exp}$ consistently serves as a critical component of the total candidate set $C$, contributing a substantial portion of the informative regions for AIMCoT.

Table 7: Proportion of salient regions selected by the AVP module of our proposed AIMCoT from the exploratory set $C_{exp}$.

| Experiment Number | 1 | 2 | 3 |
|---|---|---|---|
| M3CoT | 17.25% | 20.44% | 27.27% |
| LLaVA-W | 31.33% | 25.77% | 26.67% |

# K   DEPLOYMENT OF AIMCoT

## K.1   ANALYSIS OF THE COMPLEXITY OF AVP MODULE

**Overview of the AVP Module**   The Active Visual Probing (AVP) module's primary purpose is to dynamically and intelligently select salient sub-regions of an image during the text generation process. This is achieved by calculating the "information gain" that each potential sub-region offers, thereby allowing the model to "zoom in" on relevant visual details and generate more informed and contextually aware text.

The AVP logic is primarily encapsulated in three key methods:

1. `forward`: The main entry point where the AVP process is triggered based on changes in visual attention.
2. `_generate_candidate_regions`: Generates a diverse set of potential image regions (candidates) for evaluation.
3. `_calculate_information_gain_iterative`: The core of the AVP module. It iteratively evaluates candidate regions and selects the combination that maximizes the reduction in uncertainty (entropy) for the next token prediction.

**Definition of the Notations**   Let's define the key variables that will be used in the complexity analysis:

- $N$: The current sequence length of the input tokens.
- $N_C$: The total number of candidate regions generated (`avp_num_candidates`).
- $K$:   The   number   of   regions   to   be   selected   in   each   AVP   cycle (`avp_num_regions_to_select`).
- $G$: The grid size of the vision model's feature map (e.g., `model_vision_grid_size`, which is 4 by default, making the total number of patches $G^2 = 16$).
- $V_{sub}$: The number of visual tokens ("vokens") generated for a single cropped sub-image region.
- $\Delta N$: The length added per selected region, where $\Delta N = V_{sub} + 2$ (accounting for the `boi` and `eoi` tokens).
- $L$: The number of layers in the transformer model.
- $H$: The hidden size of the model.
- $V_{vocab}$: The size of the model's vocabulary.

**AVP Triggering in the `forward` Method**   The AVP mechanism is not activated on every forward pass; instead, it is triggered conditionally based on the change in attention directed towards the visual tokens. Specific to its operational process, in terms of attention calculation, the code calculates `latest_vattns`—which refers to the sum of attention scores from the last token to all visual patch tokens—and this step requires iterating through the attention matrices. Meanwhile, regarding the trigger condition, the core logic is `if delta_vattns > config['delta']`, where `delta_vattns` represents the difference between the current and previous visual attention sums. In conclusion, the cost of this trigger check per token generation is minimal; it primarily involves retrieving and summing pre-computed attention scores. The complexity is approximately $O(L \cdot N)$ to extract and sum the relevant attention weights to the $G^2$ visual patches, but this is dwarfed by the main model's complexity.

**`_generate_candidate_regions` Method**   This method generates $N_C$ candidate regions from the image's attention map, using a hybrid strategy that combines attention-based and random sampling. Specifically, for attention-based candidates, it first flattens the $G \times G$ attention map, then uses `torch.topk` to find the indices of the `avp_num_attention_based` patches with the highest attention—with the complexity of `topk` on a tensor of size $G^2$ being $O(G^2 \log(\text{avp\_num\_attention\_based}))$—and subsequently creates bounding boxes around these top patches, which is a constant time operation for each of the `avp_num_attention_based` candidates; for random candidates, it generates the remaining $N_C - \text{avp\_num\_attention\_based}$ candidates by randomly selecting coordinates, an operation with complexity $O(N_C - \text{avp\_num\_attention\_based})$. As a result, the

time complexity in this part is $O(G^2 \log(\texttt{avp\_num\_attention\_based}) + N_C)$, and since `avp_num_attention_based` is a small constant and $G^2$ is fixed (e.g., 16), this can be considered approximately $O(N_C)$.

**`_calculate_information_gain_iterative` Method**   This is the most computationally intensive part of the AVP module. It employs a greedy, iterative approach to select the $K$ best regions out of $N_C$ candidates. The method consists of an outer loop that runs $K$ times (for each region to be selected). Inside this loop, it evaluates the remaining candidates to pick the one that provides the highest immediate information gain.

Let's analyze a **single iteration** of this outer loop (e.g., the $k$-th iteration, where $k$ ranges from 0 to $K - 1$):

First, for the **initial entropy calculation**, it performs one forward pass through the base model (`self.model`) with the current sequence of tokens (which includes tokens from $k$ previously selected regions), where the sequence length at this stage is $N_k = N + k \cdot \Delta N$; with Key-Value (KV) caching from previous iterations, the cost can be incremental: $O(L \cdot \Delta N \cdot N_{k-1} \cdot H)$ for updating the cache with the last selected region's tokens, rather than a full $O(L \cdot N_k^2 \cdot H)$.

Next, for the **batch preparation for lookahead analysis**, the code iterates through the remaining $N_C - k$ candidate regions, and for each candidate, it performs cropping and vokenization (cropping the image pixels and passing them to `self.model.get_image_tokens`, which involves a forward pass through the vision encoder with complexity approximately $O(G^2)$ per crop, negligible compared to the transformer) and tensor concatenation (creating a new input sequence by appending the new vokens, with the length of this new sequence being $N_k + \Delta N$), with this loop running $N_C - k$ times.

Subsequently, for the **batch forward pass (lookahead)**, the $N_C - k$ new input sequences are padded and batched together, a single batched forward pass is performed on these $N_C - k$ sequences, the maximum sequence length in this batch is $N_k + \Delta N$, and since all lookahead sequences share the same prefix of length $N_k$, KV caching for the prefix can be reused across the batch; the complexity is then self-attention among suffix tokens: $O(L \cdot (N_C - k) \cdot \Delta N^2 \cdot H)$ and cross-attention to prefix: $O(L \cdot (N_C - k) \cdot \Delta N \cdot N_k \cdot H)$, with the dominant term (when $N_k \gg \Delta N$) being $O(L \cdot (N_C - k) \cdot \Delta N \cdot N_k \cdot H)$, which is the key optimization from batch processing and caching, reducing from quadratic to linear dependence on $N_k$.

Additionally, for the **information gain calculation**, it calculates the entropy for each of the $N_C - k$ outputs from the lookahead pass, which involves a softmax over the vocabulary and is $O((N_C - k) \cdot V_{vocab})$.

Finally, for the **selection and state update**, `torch.argmax` finds the best candidate in $O(N_C - k)$ time, and the base input sequence is updated for the next iteration.

**Overall Complexity of the Method**   We must sum the complexity over the K iterations of the outer loop. The most computationally intensive operation is the batched lookahead forward pass, which is significantly optimized through batch processing and Key-Value caching. The overall time complexity can be expressed as follows:

$$\sum_{k=0}^{K-1} O\left(L \cdot (N_C - k) \cdot \Delta N \cdot (N + k \cdot \Delta N) \cdot H\right). \tag{18}$$

This formula highlights that, thanks to KV caching, the complexity scales linearly with sequence length N—a substantial improvement over the standard quadratic dependence. While the cost is also linear with respect to the number of candidates $N_C$ and selections $K$, our framework maintains strong deployability. This is because the AVP module is highly efficient at extracting crucial visual information from the candidate set. Our empirical results demonstrate that AIMCoT achieves exceptional performance (as shown in Table 2) even when these hyper-parameters are kept at low levels (e.g., $K = 3, N_C = 6$, which is the default setting). In conclusion, we can approximate the complexity of AVP as follows:

$$O\left(K \cdot N_C \cdot L \cdot \Delta N \cdot (N + K \cdot \Delta N) \cdot H\right). \tag{19}$$

This demonstrates that **the combination of architectural optimizations (batching and KV caching) and the high extractive efficiency of AVP ensures the module's practicality, making it efficient and readily deployable in practice.** As a comparison, the attention-driven selection method within the baseline model ICoT is with a complexity of $O(logK \cdot \Delta N + L \cdot N \cdot H)$, which also scales linearly with sequence length N. Although this is acknowledged to be lower than the complexity of AVP shown in Equation 19, our empirical results detailed in Appendix K.2 suggest that AVP's average inference time is **no more than 1.36 times** that of this method, while **achieving performance far superior to it** as analyzed in Section 5.5.

### K.2 EMPIRICAL ANALYSIS OF THE DEPLOYMENT OF AIMCoT

In this section, we empirically investigate the deployability of the AIMCoT framework and the temporal overhead introduced by the AVP module. For the experimental setup, we utilized Chameleon-7B as the backbone in a 0-shot setting to compare the average inference time of AIMCoT against ICoT Gao et al. (2025). ICoT, as a key baseline model in this study, employs a Top-K strategy to simultaneously select regions with the highest attention scores for constructing the multimodal CoT, thereby expected to possess a relatively lower time complexity compared to AIMCoT. Consequently, the comparison with ICoT serves as a direct indicator of AIMCoT's deployability.

Table 8: Comparison of the average time to process each instance between AIMCoT and the baseline model (ICoT).

| Dataset | AIMCoT | ICoT |
|---------|--------|------|
| M3CoT | 13.37s | 11.62s |
| LLaVA-W | 11.65s | 8.58s |

The experimental results are presented in Table 8. Two key observations can be drawn. First, the AVP module does not introduce significant temporal costs to the AIMCoT framework, an efficiency attributable to batch processing and the KV Cache mechanism. Second, AIMCoT achieves substantially superior performance as shown in Table 2, at a time cost comparable to that of the efficient baseline, which is less than 1.36 times that of ICoT (specifically, 1.15 and 1.36 times on M3CoT and LLaVA-W benchmarks, respectively). This suggests that **our proposed AIMCoT framework achieves a favorable trade-off between performance and deployability.**

## L SENSITIVITY ANALYSIS OF $\delta$

The hyper-parameter $\delta$ within the DAT module serves as a crucial threshold to trigger the AVP module, which inserts salient visual regions to improve the construction of the multimodal CoT. In this section, we detail a sensitivity analysis of $\delta$ by adjusting $\delta$ across the range of $[0.1, 0.125, 0.15, 0.175, 0.2, 0.225]$ and examining not only (1) the performance of AIMCoT, but also (2) the number of times the AVP is triggered. The experiments are conducted under 0-shot setting on Chameleon-7B backbone and LLaVA-W benchmark. The experimental results are shown in Figure 5.

The left figure illustrates that AIMCoT exhibits limited performance when the threshold, $\delta$, is set too low. This underscores the importance of inserting visual information at critical moments: excessively frequent or inopportune visual insertions can disrupt the VLM's reasoning process, leading to suboptimal performance. As $\delta$ increases, the model's performance progressively improves, reaching its peak at $\delta = 0.2$ (our default setting), which corresponds to a ROUGE-L score of 0.2983. However, a further increase in $\delta$ results in a slight performance degradation. This highlights the criticality of visual information insertion for constructing an interleaved Chain of Thought: an overly stringent threshold excessively impedes the incorporation of visual data, preventing the AVP from supplying the model with necessary visual supplementation in a timely manner.

Conversely, the right figure demonstrates a consistent decrease in the number of times the AVP is triggered as $\delta$ is raised. This showcases the efficacy of $\delta$ as a threshold for modulating the activation frequency of the AVP.

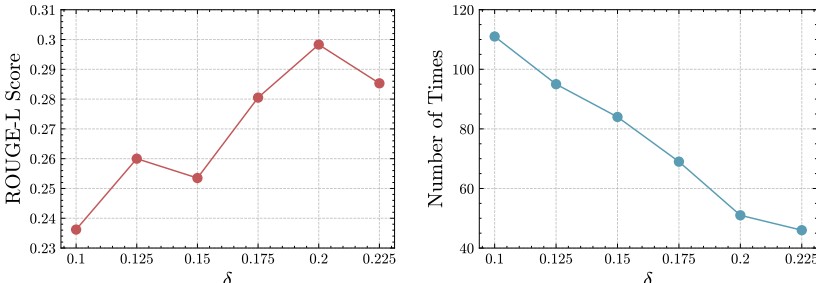

Figure 5: Experimental results of the sensitivity analysis of the hyper-parameter $\delta$. The left figure illustrates the performance of AIMCoT when $\delta$ takes different values, while the right one shows the number of times the AVP module within AIMCoT is triggered.

Table 9: Experimental results of sensitivity analysis of hyper-parameters $N_C, K$ on M3CoT benchmark, which illustrate the average time of AIMCoT to process an instance under different settings. Since $N_C, K$ inherently satisfies $N_C \geq K$, any entry corresponding to a setting that satisfies $N_C < K$ is filled with the symbol "-".

| $N_C$ | 1 | 2 | 3 | 4 | 5 | 6 | 7 |
|---|---|---|---|---|---|---|---|
| K=1 | 12.78s | 12.80s | 12.83s | 12.87s | 12.90s | 12.93s | 12.97s |
| K=3 | - | - | 13.18s | 13.23s | 13.30s | 13.37s | 13.42s |
| K=5 | - | - | - | - | 13.41s | 13.49s | 13.56s |
| K=7 | - | - | - | - | - | - | 13.66s |

Table 10: Experimental results of sensitivity analysis of hyper-parameters $N_C, K$ on LLaVA-W benchmark, which illustrate the average time of AIMCoT to process an instance under different settings. Since $N_C, K$ inherently satisfies $N_C \geq K$, any entry corresponding to a setting that satisfies $N_C < K$ is filled with the symbol "-".

| $N_C$ | 1 | 2 | 3 | 4 | 5 | 6 | 7 |
|---|---|---|---|---|---|---|---|
| K=1 | 10.61s | 10.70s | 10.75s | 10.84s | 10.92s | 11.00s | 11.08s |
| K=3 | - | - | 11.21s | 11.36s | 11.51s | 11.65s | 11.80s |
| K=5 | - | - | - | - | 11.79s | 11.96s | 12.14s |
| K=7 | - | - | - | - | - | - | 12.38s |

Table 11: Performance comparison AIMCoT variants on the basis of different constructions of the candidate set $C$.

| Construction of $C$ | M3CoT (ACC.) | LLaVA-W (ROUGE-L) |
|---|---|---|
| $C_{attn} \cup C_{rand}$   ($C_{exp} = C_{rand}$) | 31.4 | 29.8 |
| $C_{attn} \cup C_{ss}$   ($C_{exp} = C_{ss}$) | 31.2 | 29.5 |
| $C_{attn} \cup C_{fsam}$   ($C_{exp} = C_{fsam}$) | 31.0 | 29.6 |
| $C_{attn}$ | 30.8 | 28.9 |
| $C_{rand}$ | 30.4 | 28.6 |
| $C_{ss}$ | 30.3 | 28.7 |
| $C_{fsam}$ | 29.9 | 27.7 |

## M   SENSITIVITY ANALYSIS OF $K, N_C$

Our proposed AIMCoT incorporates the design of AVP module. In contrast to existing research, AIMCoT, benefiting from the AVP module, does not simply select the top-$K$ regions with the highest attention scores from the attention map. Instead, it meticulously selects $K$ regions from a total set $C$ of $N_C$ candidate regions to construct the multimodal CoT ($N_C = N + M$). However, this approach may inevitably raise concerns regarding the deployability of AIMCoT, particularly as the hyper-parameters $N_C$ and $K$ increase.

To investigate this, we conduct experiments to explore the average processing time per instance for AIMCoT with larger values of $N_C$ and $K$. For the experimental setup, we implement AIMCoT with the Chameleon-7B backbone on the M3CoT and LLaVA-W datasets under various combinations of $N_C$ and $K$. The results are presented in Tables 9 and 10, from which we derive two key insights:

- **Insensitivity to the growth of the candidate set size** $N_C$: observing any row with a fixed $K$ (e.g., $K = 3$ on Table 9), as $N_C$ increases from 3 to 7, the total processing time rises from 13.18s to 13.42s, a marginal increase of only 0.24s. This implies that each additional candidate region introduces an average overhead of less than 0.06s. This strongly demonstrates that the performance of the AVP module does not degrade sharply with a moderate expansion of the candidate pool, indicating excellent scalability.

- **Diminishing marginal cost with the increase in the number of selections** $K$: considering a fixed column for $N_C$ (e.g., $N_C = 7$ on Table 10), as $K$ increases from $1 \rightarrow 3$, $3 \rightarrow 5$, and $5 \rightarrow 7$, the processing time increases by 0.72s, 0.34s, and 0.24s, respectively, which shows a notable decrease in incremental cost. This suggests that the primary computational overhead of the AVP algorithm lies in the initiation of the iterative search (the jump from $K = 1$ to $K = 3$). Once the iteration begins, the cost of subsequent selection steps is remarkably low, benefiting from efficient mechanisms such as the batch processing and KV Cache.

Based on these key insights, our proposed AVP module demonstrates high computational efficiency and robustness when faced with increased computational complexity (i.e., larger $N_C$ and $K$ values).

## N   ABLATION STUDY ON THE CONSTRUCTION OF CANDIDATE SET $C$

In this section, we investigate the influence of different compositions of the total set $C$ on the performance of our proposed AIMCoT. We specifically examine two primary configurations:

**Constructing $C$ using only $C_{attn}$ or $C_{exp}$.** For the latter, we evaluate four distinct construction methodologies for $C_{exp}$: (a) $C_{rand}$: uniform random sampling; (b) $C_{ss}$: the selective search algorithm Uijlings et al. (2013), which is the seminal region proposal method utilized in R-CNN Girshick et al. (2014); (c) $C_{fsam}$: FastSAM Zhao et al. (2023), a computationally efficient variant of the foundational vision segmentation model, SAM Kirillov et al. (2023).

**Constructing $C$ using both $C_{attn}$ and $C_{exp}$.** Similarly, as for $C_{exp}$, we also consider its diversified construction, including $C_{rand}, C_{ss}$, and $C_{fsam}$.

It is worth noting that although the construction is diverse, the size of $C$ remains consistent. When $C$ is composed of $C_{attn}$ and $C_{exp}$, the two each account for half. The experimental results are shown in Table 11.

As observed, the combination of the two sets (i.e., $C = C_{attn} \cup C_{exp}$) invariably yields superior performance for AIMCoT compared to configurations where either $C = C_{attn}$ or $C = C_{exp}$ is used exclusively. This highlights the importance of diversifying the sources of candidate visual regions.

When comparing the different construction methods for $C_{exp}$, the performance gap among models is marginal when used in conjunction with $C_{attn}$. Specifically, despite its simplicity, random sampling achieves highly competitive results, which motivates our choice to adopt it as the default method for constructing $C_{exp}$. Intuitively, the advantage of random sampling lies in its ability to provide regions across different parts of the image unbiasedly with maximal spatial diversity.

