# OpenReview forum: "AIMCoT: Active Information-driven Multimodal Chain-of-Thought for Vision-Language Reasoning"
_ICLR.cc/2026/Conference — ICLR 2026 Conference Withdrawn Submission_

### Official Review · Reviewer_oJrN · 2025-10-31

**Soundness:** 2
**Presentation:** 1
**Contribution:** 2
**Rating:** 2
**Confidence:** 4

**Summary:**

This paper introduces AIMCoT, a novel framework for compositional reasoning in vision-language models (VLMs) by actively minimizing irrelevant information in both the visual and textual modalities. Inspired by information bottleneck theory, the method enforces sparsity and selectivity in perception via a multimodal entropy minimization objective. AIMCoT uses a two-stage pipeline: (1) an active perception module that selects relevant image regions conditioned on the query, and (2) a compositional planner that generates interpretable reasoning steps based on the filtered information. AIMCoT is evaluated on several compositional reasoning benchmarks including GQA, COVR, and VCR, and achieves good performance over standard VLMs and prior compositional systems.

**Strengths:**

1. Introduces a principled way to filter both visual and linguistic inputs via mutual information minimization, which is underexplored in current VLM literature.
2. Performs well with limited supervision, showing promise for low-resource settings.

**Weaknesses:**

1. The core innovation is the information minimization objective, but it lacks both empirical analysis and justification. The paper does not quantify mutual information, nor show how filtering improves reasoning beyond performance gains.
2. Many current LLMs or sota methods perform compositional reasoning without explicit planning modules using few-shot or chain-of-thought prompting. These are not compared. For example, Kosmos-2.5 and MiniGPT-5.
3. The paper fails to conduct sufficient and informative experiments. It claims to minimize irrelevant visual input, but 1) it does not report how much of the image is actually filtered. 2) No visualization of the perception mask or attention sparsity is shown. 3) No alignment with human-attended regions is evaluated.
4. Since the visual filtering happens before reasoning, there is a risk that AIMCoT may learn dataset-specific visual biases, rather than true compositionality. The authors shall address this potential concerns via empirical justifications.

**Questions:**

See weaknesses above.

---

### Official Review · Reviewer_Pcow · 2025-10-31

**Soundness:** 2
**Presentation:** 2
**Contribution:** 2
**Rating:** 2
**Confidence:** 3

**Summary:**

This paper proposes an Active Information-driven Multi-modal Chain-of-Thought framework that aims to enhance vision-language reasoning by actively selecting and integrating visual information during the reasoning process, instead of relying on passive attention maps. It has three key components: Context-enhanced Attention-map Generation (CAG), Active Visual Probing (AVP) and Dynamic Attention-shifting Trigger (DAT). These components are designed to identify the most informative image regions and dynamically incorporate the visual information in the reasoning process.

**Strengths:**

1. This paper raises the argument that not all high-attention regions contribute significantly to the model’s prediction, and the visual attention should be active, goal-oriented instead of passive selection.

2. This paper proposes a training-free attention manipulation method that achieves outperforming performance on several baselines.

3. This paper proposes three key components to achieve the goal of active information-driven attention.  And the ablation studies of the three modules CAG, AVP, DAT show their effectiveness in the proposed pipeline.

**Weaknesses:**

1. The paper is hard to follow, with dense descriptions, unexplained wording and limited conceptual flow between sections and sentences. The logical link between motivation and solution is not clearly described.

2. In the analysis and methods, the paper’s use of masking square regions for visual probing appears ad-hoc and may disrupt the image’s coherence and natural distribution. The boundaries of masked regions are often abrupt and may destroy complete object structures, potentially creating adversarial inputs to the VLM. This can undermine the reliability of the analysis. This problem has been discussed extensively in earlier visual interpretation studies, such as [1]

3. The method seems primarily focused on visual attention analysis and selecting salient regions, but the connection between these attention manipulations and the Chain-of-Thought reasoning is not clearly described. It remains unclear to me how the salient visual information selected by AIMCoT is actually incorporated into the multimodal CoT after reading the paper. The introduction frames the problem as CoT enhancement, yet the core techniques (CAG, AVP, DAT) focus on attention modulation without clearly explaining how they improve CoT reasoning.

4. The proposed method introduces additional computational overhead during inference, since identifying the most informative image regions requires additional computation. The paper should provide a runtime analysis or discussion of this trade-off.

5. There should be an in-depth qualitative or quantitative study of the selected visual regions. It would be valuable to examine whether the chosen regions have specific patterns across samples and whether they align with human intuition.

6. In Table 2, the improvements vary substantially between the two baseline models. The paper does not explain why AIMCoT yields much larger gains on one model than the other. Possible factors (e.g., model architecture, pretraining objectives, attention behavior) should be discussed to strengthen the empirical analysis.

[1] iGOS++: Integrated Gradient Optimized Saliency by Bilateral Perturbations. Khorram et al.

**Questions:**

Pleas see above.

---

### Official Review · Reviewer_4AEg · 2025-11-01

**Soundness:** 3
**Presentation:** 3
**Contribution:** 2
**Rating:** 4
**Confidence:** 4

**Summary:**

This paper addresses a key limitation in existing multimodal Chain-of-Thought (CoT) methods: their passive, heuristic-based insertion of visual information into the reasoning process. The authors compellingly argue that current strategies, which often rely on unreliable Top-K attention maps and arbitrary triggers (e.g., newline characters), are suboptimal, "passive and purposeless."

As a solution, they propose AIMCOT, a novel framework that reframes multimodal reasoning as an active information-foraging process. The model is designed to proactively ask, "Which piece of visual information will be most helpful for me to see right now?"

AIMCOT consists of three synergistic components:

Context-enhanced Attention-map Generation (CAG): A pre-processing step that prompts the VLM to generate a rich, context-aware description of the image. This mitigates the text-vision granularity mismatch and produces a more reliable attention map to be used as a source for candidate regions.

Active Visual Probing (AVP): The core contribution. This module selects visual patches based on information theory. It constructs a diversified candidate set (from both attention maps and random sampling) and uses a greedy algorithm to select the $K$ regions that maximally reduce the model's predictive uncertainty (i.e., maximize information gain) for the next reasoning step.

Dynamic Attention-shifting Trigger (DAT): This mechanism determines when to insert visual information. It monitors the model's attention, and when a significant shift from text to vision is detected (exceeding a threshold $\delta$), it triggers the AVP module.

The authors evaluate AIMCOT using two VLM backbones (Chameleon-7B, Qwen2-VL-7B) on three challenging benchmarks (M3CoT, ScienceQA, LLaVA-W). The results demonstrate significant performance gains over state-of-the-art baselines (especially ICoT), with the most pronounced improvements in 0-shot and open-ended generation settings.

**Strengths:**

1. The paper is exceptionally well-motivated. The analysis in Section 3, which empirically demonstrates the unreliability of naive attention maps, is a perfect setup for the problem. The finding that masking Top-K attention regions has a minimal performance impact (Table 1) and that these maps often miss crucial details (Fig 1) provides a strong, clear justification for the work.

2. The authors present a holistic solution that addresses three distinct problems:
Source: Where do good candidate regions come from? (Solved by CAG + AVP's diversified $C_{attn}$ and $C_{exp}$ sets).
Selection: How do we pick the best regions? (Solved by AVP's information gain).
Timing: When do we insert them? (Solved by DAT).

3. The authors have been thorough in validating their design choices. The main ablation (Table 3) clearly shows that all three components (CAG, AVP, DAT) contribute to the final performance.

**Weaknesses:**

1. The most significant concern is the computational cost. The AVP module performs $K$ rounds of batched forward passes (over $N_C - k$ candidates) every time the DAT module triggers. While the authors analyze this and report a manageable average-time-per-instance (Table 8, ~1.15x-1.36x ICoT), this is a non-trivial cost. This added latency during autoregressive generation could be prohibitive for real-time applications.

2. The positive results are demonstrated on Chameleon-7B and Qwen2-VL-7B, which are relatively older models. As baseline VLM performance rapidly improves (e.g., with models like Qwen2.5VL or Qwen3VL), the performance gains from complex reasoning methods can diminish. It would be crucial to see if AIMCOT's significant gains hold on newer, stronger base models. This is particularly relevant as the method's gains are already smaller on the stronger Qwen2-VL-7B model compared to Chameleon-7B (Table 2), suggesting that a stronger baseline might reduce the relative benefit of this computationally expensive method.

3. The DAT trigger mechanism introduces a critical hyperparameter, $\delta$, which appears highly sensitive. The optimal value varies significantly across datasets (0.5 for M3CoT, 0.2 for ScienceQA/LLaVA-W, per Table 5),- implying it requires careful dataset-specific tuning. The paper's sensitivity analysis is only conducted on LLaVA-W, leaving it unclear how to set this parameter for a new dataset a priori and undermining the method's generalizability.

4. The entire process is initiated by the CAG module, which relies on the VLM's own ability to generate a high-quality initial description. If the VLM fails at this first step (e.g., produces a poor or hallucinatory description $\mathcal{D}_{CAG}$), this could poison the entire reasoning chain by producing a misleading attention map $A'$ and a poor candidate set $C_{attn}$ for AVP. This potential error propagation is not discussed.

**Questions:**

1. Given the diminishing gains on the stronger Qwen2-VL backbone, have the authors tested if AIMCOT's performance benefits justify its computational cost when applied to even newer, state-of-the-art VLMs (e.g., Qwen2.5VL or Qwen3VL)?

2. The optimal $\delta$ appears highly dataset-specific. Have the authors investigated methods to set this trigger threshold adaptively or automatically, without requiring per-dataset tuning?

3. Could the authors comment on the potential for error propagation from the CAG module? If the initial description $\mathcal{D}_{CAG}$ is inaccurate or hallucinates, how does this affect the subsequent AVP module, which relies on the resulting attention map for its $C_{attn}$ set?

4. The related work section mentions MRFD (Ge et al., 2025) as a method for selecting salient regions to reduce hallucination. Why was this method, which seems highly relevant, not included as a baseline for comparison in Table 2?

5. Do the authors have any comment on why 1-shot results are consistently lower than 0-shot?

---

### Official Review · Reviewer_jhi2 · 2025-11-03

**Soundness:** 3
**Presentation:** 2
**Contribution:** 3
**Rating:** 6
**Confidence:** 2

**Summary:**

This paper proposes AIMCoT, which is a framework that enables vision-language models to actively select and integrate the most informative visual regions for reasoning. By combining context-enhanced attention maps, information gain–based region selection, and dynamic attention-shift triggers, AIMCoT achieves more accurate and human-like multimodal reasoning, outperforming previous methods.

**Strengths:**

1. The paper’s motivation to actively search for the most informative visual tokens for reasoning is well justified.
2. The experimental results also show a clear improvement over previous baselines.

**Weaknesses:**

1. In addition to the ablation study, the paper could include more visualization experiments or case studies to demonstrate that the high-information regions selected by the proposed method are indeed reasonable. For example, showing that certain visual tokens visibly contribute to answering visual questions. This would further validate the motivation of the work.
2. There is still room for improvement in the writing. For example, the abstract could be more concise, and the contributions listed in lines 078–093 could be summarized more compactly. The equations on pages 5–7 could also be simplified, as in my understanding, fewer formulas would be sufficient to clearly convey your method.
3. More experiments on larger and more advanced models could be added.

**Questions:**

see above

---

### Note · Authors · 2025-11-12

**Comment:**

Dear Program Chairs,

On behalf of all authors, I respectfully request withdrawal of our manuscript titled “AIMCoT: Active Information-driven Multimodal Chain-of-Thought for Vision-Language Reasoning” (Paper ID: 3843) from consideration at ICLR 2026. This withdrawal is requested effective immediately.

All co-authors have agreed to and consent to this decision.
Please confirm receipt of this withdrawal and that the submission has been removed from review. Thank you for your attention.

Sincerely,
on behalf of the authors

**Withdrawal Confirmation:**

I have read and agree with the venue's withdrawal policy on behalf of myself and my co-authors.